# GNN-LM: Language Modeling based on Global Contexts via GNN

**Yuxian Meng[1], Shi Zong[2], Xiaoya Li[1], Xiaofei Sun[1,4], Tianwei Zhang[3], Fei Wu[4], Jiwei Li[1,4]**
[1]Shannon.AI, [2]Nanjing University,[3]Nanyang Technological University, [4]Zhejiang University
{yuxian_meng, xiaoya_li, xiaofei_sun, jiwei_li}@shannonai.com, szong@nju.edu.cn
tianwei.zhang@ntu.edu.sg, wufei@zju.edu.cn

## Abstract

Inspired by the notion that "*to copy is easier than to memorize*", in this work, we introduce GNN-LM, which extends vanilla neural language model (LM) by allowing to reference similar contexts in the entire training corpus. We build a directed heterogeneous graph between an input context and its semantically related neighbors selected from the training corpus, where nodes are tokens in the input context and retrieved neighbor contexts, and edges represent connections between nodes. Graph neural networks (GNNs) are constructed upon the graph to aggregate information from similar contexts to decode the token. This learning paradigm provides direct access to the reference contexts and helps improve a model's generalization ability. We conduct comprehensive experiments to validate the effectiveness of the GNN-LM: GNN-LM achieves a new state-of-the-art perplexity of 14.8 on WikiText-103 (a 3.9 point improvement over its counterpart of the vanilla LM model), and shows substantial improvement on One Billion Word and Enwiki8 datasets against strong baselines. In-depth ablation studies are performed to understand the mechanics of GNN-LM.[1]

## 1 Introduction

Language modeling (LM) is a basic and long-standing task in natural language processing (Shannon, 2001; Bahl et al., 1983; Chen & Goodman, 1999; Mikolov et al., 2012; Xie et al., 2017). It aims at predicting the upcoming token given the sequence of previous context consisting of a sequence of tokens. A common practice to train a language model is to enforce the model to maximize the probability of the upcoming ground-truth token at training time. At test time, the next token to predict could be the one with the highest probability (via greedy search) or the one that maximizes a window of tokens through the beam search strategy. This form of training-test procedure can be viewed as a process of *memorization*, or doing a *close-book examination*, if we compare the training data to a book and inference to doing an examination: The process of iterating $N$ epochs over the training data is comparable to reviewing the book $N$ times and the model needs to memorize what is the most likely to appear given specific context based on the training data. At test time, the book needs to be closed, i.e., the model does not have means to refer to the training data at test time, and the model has to invoke related memory to predict the next token during inference.

There are two limitations to this *close-book examination* strategy: (1) the memorization-based language models are usually hard to memorize the knowledge of hard examples (e.g., long-tail cases in the training set); (2) memory required to memorize the whole training data is usually intensive. The difficulty of resolving these two problems can be substantially alleviated if the model can be provided with related contexts from the training set so that the model can reference them for decisions. This process can be viewed as a strategy different from memorization or *close-book examination – copy*, or in other words, *open-book examination*. For example, given a prefix "*J. K. Rowling is best known for writing*" and we want to predict the upcoming token, a language model will more easily generate token "*Harry*" if it can refer to the context "*J. K. Rowling wrote the Harry Potter* fantasy series".

Motivated by the observation that "*to copy is easier than to memorize*", or "*an open-book exam is easier than a close-book exam*", in this work, we introduce a new language modeling scheme –

---

[1]The code can be found at `https://github.com/ShannonAI/GNN-LM`

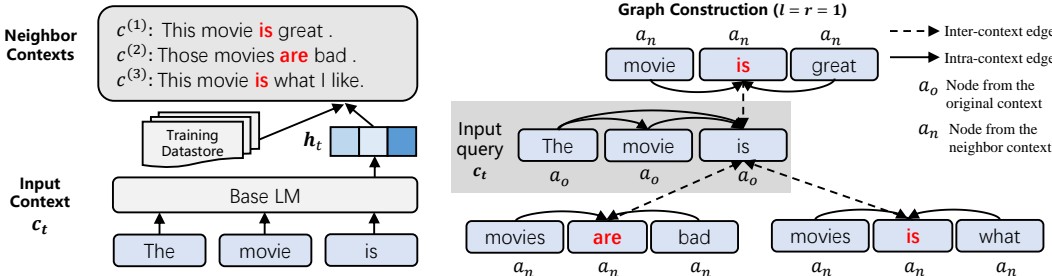

Figure 1: An overview of the proposed GNN-LM model pipeline. *Left*: Given an input context $c_t = (w_1, \cdots, w_{t-1})$ (here the context is "*The movie is*"), a base LM model encodes it into a high-dimensional representation $h_t$, which is then used to query the training datastore to retrieve the nearest contexts along with the visited tokens (marked in red). *Right*: The tokens in the input context and the retrieved tokens comprise a graph and are viewed as two types of nodes: nodes from the original text and nodes from the neighbor text. Intra-context edges link tokens within the same input, and inter-context edges link tokens from the retrieved contexts to the original context. After modeling the graph as a whole with GNNs, we use the updated representation of $w_{t-1}$ (token "*is*" in this example) to compute the likelihood of the next token.

GNN-LM, which provides an LM model with the ability to reference similar contexts from the entire training corpus as cues for prediction. The similar contexts, defined as the $k$ neighbors of the input in the training corpus, are served as additional references for the model to predict the next token. To integrate retrieved neighbors with the input, we build a directed heterogeneous graph on top of the input and the extracted contexts, where nodes are the tokens and edges represent the connections between them. We define two types of nodes – the original node from the input context and the neighbor node from the extracted contexts, and two types of edges – the inter-context edge and the intra-context edge that respectively associate inter (i.e., between retrieved contexts and input context) and intra (i.e., context within the input) contexts. A graph neural network (GNN) is employed to aggregate information from both inter-context and intra-context, which is used to generate the target token. We observe that the proposed scheme retrieves the related contexts as references, making it significantly easier for the model to predict upcoming words in the LM task.

We further combine GNN-LM with $k$NN-LM (Khandelwal et al., 2019), an orthogonal technique enhancing language models, to improve the overall performance of our model. We carry out experiments on three widely used language modeling benchmarks: WikiText-103, One Billion Word and Enwik8. Experimental results show that our proposed framework outperforms the strong baseline on all three benchmarks. Specifically, applying the GNN-LM framework to a strong base LM leads to a substantial performance boost (-1.9 perplexity) on WikiText-103, and combining with $k$NN-LM achieves a new state-of-the-art perplexity of 14.8 – a 3.9 point improvement over the base LM. We perform comprehensive analyses including complexity analysis and the effects of different components to better understand the mechanics of GNN-LM.

## 2 GNN-LM

### 2.1 OVERALL PIPELINE

We present the overall pipeline of our model in Figure 1. At each time step $t$, a neural language model (LM) $f(\cdot)$ first encodes a sequence of context tokens $c_t = (w_1, w_2, ..., w_{t-1})$ to a high-dimensional representation $h_t = f(c_t) \in \mathbb{R}^d$, where $d$ is the dimension of hidden states. Then a transformation matrix $W \in \mathbb{R}^{V \times d}$ is used to estimate the probability of the $t$-th token $p(w_t|c_t) = \text{softmax}(Wh_t)$, where $V$ is the size of the vocabulary. We augment the vanilla neural language model by allowing it to reference samples in the training set that are similar to the current decoded sequence. Concretely, we leverage a novel self-attention augmented Graph Neural Network (GNN) on top of the vanilla LM to enable message passing between the context $c$ and retrieved reference tokens from the training set, updating the representation $h_t$ generated by the vanilla LM. The updated representation, which aggregates additional information from reference tokens, is then used to estimate $p_{\text{LM}}(w_t|c_t)$.

## 2.2 Graph Construction

The first step of our proposed framework is to build a graph capturing the connections between the context tokens $c_t = (w_1, w_2, ..., w_{t-1})$ and those similar to $c_t$ in the training set. To this end, we construct a directed heterogeneous graph, where the nodes are tokens from $c_t$ or the tokens from the neighbor contexts retrieved from the training set, and the edges represent different relationships between the nodes to be discussed below.

Formally, we define a graph as $\mathcal{G} = (\mathcal{V}, \mathcal{E}, \mathcal{A}, \mathcal{R}, \tau, \phi)$, where $\mathcal{V}$ is a collection of nodes $v$ and $\mathcal{E}$ is a collection of edges $e$. We define two types of nodes $\mathcal{A} = \{a_o, a_n\}$, where $a_o$ means that the node is within the input $c_t$. $a_n$ means the node is in $\mathcal{N}(c_t)$, the set of extracted contexts within the neighborhood of $c_t$. We also define two types of edges $\mathcal{R} = \{r_{\text{inter}}, r_{\text{intra}}\}$, where $r_{\text{inter}}$ means inter-context connection (from $a_n$ nodes to $a_o$ nodes) and $r_{\text{intra}}$ means intra-context connection (between two nodes of same type). Each token within the input is a node of type $a_o$, and edges of type $r_{\text{intra}}$ are constructed from node $w_i$ to $w_j$ ($i \leq j$), which can be viewed as a graph interpretation of the transformer structure. Both nodes and edges are associated with their respective type mapping functions $\tau(v) : \mathcal{V} \to \mathcal{A}$ and $\phi(e) : \mathcal{E} \to \mathcal{R}$.

For an input context $c_t$, we retrieve $k$ nearest neighbors $\mathcal{N}(c_t) = \{c_{t_1}^{(1)}, ..., c_{t_k}^{(k)}\}$ of $c_t$ from the training set as follows: we first use $h_t$ to query the cached representations of all tokens for training samples, where the cached representations are obtained by a pretrained LM. The distance is measured by the cosine similarity,[2] and we retrieve the top $K$ tokens denoted by $\{w_j^{(i)}\}$. The superscript $^{(i)}$ denotes the $i$-th training sample and the subscript $_j$ denotes the $j$-th time step. $w_j^{(i)}$ thus means that the $j$-th time step of the $i$-th training sample is retrieved as one of the nearest neighbors to $h_t$. $w_j^{(i)}$ is expanded to $c_j^{(i)}$ by adding both left and right contexts, where $c_j^{(i)} = \{w_{j+p}^{(i)}\}_{p=-l}^r$, where $l$ and $r$ respectively denote the left and right window size. The corresponding representations $\{h_{j+p}^{(i)}\}_{p=-l}^r$ are used as the initialized node embeddings.

Different from $k$NN-LM (Khandelwal et al., 2019) that uses $w_{j+1}^{(i)}$, which is the token right after the retrieved token $w_j^{(i)}$, to directly augment the output probability, we explicitly take advantage of all contextual tokens near $w_{t_i}^{(i)}$ as additional information in the form of graph nodes. In this way, the model is able to reference similar *contexts* in the training set and leverage the corresponding ground-truth target tokens via the heterogeneous graph built on both the original input tokens and the context reference tokens.

For the neighbor context window size $l$ and $r$, we set $l = r = 1$ in all experiments. During experiments, we find that using shallow (i.e., 3) GNN layers and adding $r_{\text{intra}}$ edges between adjacent tokens can alleviate overfitting. Since a 3-layer GNN only aggregates information from 3-hop nodes in the graph, using larger $l$ and $r$ have no influence on GNN representations.

## 2.3 GNN on the Constructed Graph

We now use graph neural networks (GNNs) to aggregate and percolate the token information based on the graph constructed in Section 2.2. In this work, to accommodate the modeling of $r_{\text{intra}}$ from node $w_i$ to $w_j$ ($i \leq j$) within $c_t$, where Transformer with self-attention is usually adopted, we extend the self-attention mechanism to $r_{\text{inter}}$, and construct a self-attention augmented GNN.

Specifically, the $l$-th layer representation of node $n$ is computed by (here we use the superscript $^{[l]}$ to represent the $l$-th layer):

$$h_n^{[l]} = \underset{\forall s \in \mathcal{N}(n)}{\text{Aggregate}}(\text{Attention}(s, e, n) \cdot \text{Feature}(s, e, n)) + h_n^{[l-1]}. \tag{1}$$

$\text{Attention}(s, e, n)$ estimates the importance of the source node $s$ on target node $n$ with relationship $e$, $\text{Feature}(s, e, n)$ is the information feature that $s$ should pass to $n$, and $\text{Aggregate}(\cdot)$ aggregates the neighborhood message with the attention weights. To draw on the information in the heterogeneous graph, we use different sets of parameters for different node types $\tau(\cdot)$ and different edge types $\phi(\cdot)$ akin to Hu et al. (2020).

---

[2]In practice, we use FAISS (Johnson et al., 2019) for fast approximate $k$NN search.

**Attention**  Similar to the multi-head attention mechanism of Transformer (Vaswani et al., 2017), the Attention$(\cdot, \cdot, \cdot)$ operator in our model consists of $h$ heads, which compute attention weights independently, followed by concatenation to get the final output. For simplicity, we only describe the single-head situation below. For each edge $(s, e, n)$, the representation of target node $n$ is mapped to a query vector $Q(n)$, and the representation of source node $s$ is mapped to a key vector $K(s)$. The scaled inner-production is then used to compute the attention weight between $Q(n)$ and $K(s)$, which is further normalized over all edges that have the same edge type:

$$K(s) = \boldsymbol{W}^k_{\tau(s)} \boldsymbol{h}^{[l-1]}_s, \quad Q(n) = \boldsymbol{W}^q_{\tau(n)} \boldsymbol{h}^{[l-1]}_n,$$

$$\text{Attention}(s, e, n) = \frac{1}{Z} \exp\left( K(s) \boldsymbol{W}^{\text{ATT}}_{\phi(e)} Q(n)^\top \cdot \frac{\boldsymbol{\mu}_{\langle \tau(s), \phi(e), \tau(n) \rangle}}{\sqrt{d}} \right), \tag{2}$$

$$Z = \sum_{s' \in \mathcal{N}(n), e' \in \phi(e)} \text{Attention}(s', e', n),$$

where $d$ is the hidden dimensionality, and $\boldsymbol{W}^q_{\tau(s)} \in \mathbb{R}^{d \times d}$, $\boldsymbol{W}^k_{\tau(n)} \in \mathbb{R}^{d \times d}$, $\boldsymbol{W}^{\text{ATT}}_{\phi(e)} \in \mathbb{R}^{d \times d}$, $\boldsymbol{\mu} \in \mathbb{R}^{|\mathcal{A}| \times |\mathcal{R}| \times |\mathcal{A}|}$ are learnable model parameters.

**Feature**  Parallel to the calculation of attention weights, we propagate information from source node $s$ to target node $n$. The single-head feature is defined by:

$$\text{Feature}(s, e, n) = \boldsymbol{W}^v_{\tau(s)} \boldsymbol{h}^{[l-1]}_s \boldsymbol{W}^{\text{FEA}}_{\phi(e)}, \tag{3}$$

where $\boldsymbol{W}^v_{\tau(s)} \in \mathbb{R}^{d \times d}$ and $\boldsymbol{W}^{\text{FEA}}_{\phi(e)} \in \mathbb{R}^{d \times d}$ are learnable model parameters.

**Aggregate**  Aggregate$(\cdot)$ weight-sums the feature Message$(s, e, n)$ within the vicinity using Attention$(s, e, n)$, and the result is then linearly projected into a $d$-dimensional representation:

$$\text{Aggregate}(\cdot) = \boldsymbol{W}^o_{\tau(n)} \left( \underset{\forall s \in \mathcal{N}(n)}{\oplus} \left( \text{Attention}(s, e, n) \cdot \text{Feature}(s, e, n) \right) \right) \tag{4}$$

where $\oplus$ is element-wise addition and $\boldsymbol{W}^o_{\tau(n)} \in \mathbb{R}^{d \times d}$ is model parameter. The representation of token $w_{t-1}$ from the last layer is used to compute the language model probability $p_{\text{LM}}(w_t | \boldsymbol{c}_t)$.

### 2.4 $k$NN Based Probability for Next Token

We further incorporate the proposed model with $k$NN (Khandelwal et al., 2019; 2020; Meng et al., 2021), a related but orthogonal technique, to improve the performance of our model. It extends a vanilla LM by linearly interpolating it with a $k$-nearest neighbors (kNN) model. Concretely, for each input context $\boldsymbol{c}_t = (w_1, w_2, ..., w_{t-1})$, we retrieve the $k$ nearest neighbors $\mathcal{N}(\boldsymbol{c}_t) = \{\boldsymbol{c}^{(1)}_{t_1}, ..., \boldsymbol{c}^{(k)}_{t_k}\}$, and compute the $k$NN based probability for the next token by:

$$p(w_t | \boldsymbol{c}_t) = \lambda p_{\text{kNN}}(w_t | \boldsymbol{c}_t) + (1 - \lambda) p_{\text{LM}}(w_t | \boldsymbol{c}_t),$$

$$p_{\text{kNN}}(w_t | \boldsymbol{c}_t) = \frac{1}{Z} \sum_{i=1}^{k} \mathbb{1}_{w_t = w^{(i)}_{t_i}} \exp\left( \cos(f(\boldsymbol{c}_t), f(\boldsymbol{c}^{(i)}_{t_i})) / T \right), \tag{5}$$

with $Z$ being the normalization factor, $f(\cdot)$ is the neural language model encoding contexts to high dimensional representations, $\cos(\cdot, \cdot)$ is cosine similarity, and $\lambda$ and $T$ are hyperparameters.[3]

## 3 Experiments

We conduct experiments on three widely-used language modeling datasets: WikiText-103 (Merity et al., 2016), One Billion Word (Chelba et al., 2013) and Enwik8 (Mahoney, 2011). For all experiments, we add a 3-layer self-attention augmented GNN on top of the pretrained base LM, and use

---

[3]The original version of $k$NN-LM (Khandelwal et al., 2019) uses negative $L_2$ distance as vector similarity, and does not have hyperparameter $T$. We followed Khandelwal et al. (2020) to add hyperparameter $T$ and followed Meng et al. (2021) to use cosine similarity.

the same hidden dimension and number of heads as our base LM. We retrieve $k = 1,024$ nearest neighbors for each source token, among them the top 128 neighbors are used in graph, and all of them are used in computing the $k$NN-based probability $p_{kNN}(w_t|\boldsymbol{c}_t)$. For the neighbor context window size $l$ and $r$ in Section 2.2, we set $l = 1$ and $r = 1$.

## 3.1 TRAINING DETAILS

**KNN Retrieval**   In order to reduce memory usage and time complexity, in practice we use FAISS (Johnson et al., 2019) for fast approximate $k$NN search. Concretely, we quantized each dense vector to $q$ bytes, followed with a clustering of all vectors to $C$ clusters. During retrieval, we only search in 32 clusters whose centroids are nearest to query vector. For WikiText-103 and Enwik8 datasets, which contain approximately 100M tokens, we set $q = 128$ and $c = 4,096$. For One Billion Word dataset, we set $q = 64$ and $c = 1,048,576\,(2^{20})$ for faster search.

**Data Leakage Prevention**   When searching for the $k$ nearest neighbors of $\boldsymbol{c}_t = (w_1, w_2, ..., w_{t-1})$, we need to make sure each reference neighbor token does not leak information for $w_t$. Specifically, we should not retrieve $\boldsymbol{c}_{t+1} = (w_1, w_2, ..., w_t)$ as reference, otherwise the model prediction is trivial to optimize since the information of target token is already included in the graph. Let $T$ be the maximum sequence length and $L$ be the number of layers. Practically, the representation of each token is dependent on previous $T$ and $T \times L$ tokens for Transformer and Transformer-XL, respectively. Therefore we ignore all the neighboring nodes within this interval in graph construction during training. During inference, we do not impose this constraint.

**Feature Quantization**   The input node representations of the graph neural network $H^{[0]}$ are generated by a pretrained neural language model. To accelerate training and inference, we wish to cache all token representations of the entire training set. However, frequently accessing Terabytes of data is prohibitively slow. To address this issue, we followed Meng et al. (2021) to use product quantization (PQ) (Jegou et al., 2010; Ge et al., 2013) to compress the high-dimensional representation of each token. In our experiments, quantizing representations from 1,024-dimension floating-point dense vectors to 128 bytes reduces the memory consumption from 2.3TB to 96GB for the One Billion Word dataset, thus making the end-to-end model training feasible.

## 3.2 MAIN RESULTS

**WikiText-103**   WikiText-103 is the largest available word-level language modeling benchmark with long-term dependency. It contains 103M training tokens from 28K articles, and has a vocabulary of around 260K. We use the base version of deep Transformer language model with adaptive embeddings (Baevski & Auli, 2018) as our base LM. This model has 16 decoder layers. The dimensionality of word representations is 1,024, the number of multi-attention heads is 16, and the inner dimensionality of feedforward layers is 4,096. During training, data is partitioned into blocks of 3,072 contiguous tokens. During evaluation, blocks are complete sentences totaling up to 3,072 tokens of which the first 2,560 tokens serve as context to predict the last 512 tokens. As shown in Table 1, GNN-LM reduces the base LM perplexity from 18.7 to 16.8, which demonstrates the effectiveness of the GNN-LM architecture. The combination of GNN and $k$NN further boosts the performance to 14.8, a new state-of-the-art result on WikiText-103.

**One Billion Word**   One Billion Word is a large-scale word-level language modeling dataset of short-term dependency. It does not preserve the order of sentences, contains around 768M training tokens and has a vocabulary of around 800k. We adopt the very large version of Transformer model in Baevski & Auli (2018) as our base LM. Results in Table 2 show that GNN-$k$NN-LM helps base LM reduce 0.5 perplexity with only 27M additional parameters. For comparison, Baevski & Auli (2018) use 560M additional parameters to reduce perplexity from 23.9 to 23.0.

**Enwik8**   Enwik8 is a character-level language modeling benchmark that consists of 100M characters from English Wikipedia articles, and has a vocabulary of 208. For base LM, we use Transformer-XL (Dai et al., 2019) with 12 layers, 8 heads, 512 dimensional embedding and 2,048 dimensional inner feed forward layer. Table 3 shows that GNN-$k$NN-LM outperforms base LM by 0.03 Bit per

| Model | # Param | Test ppl ($\downarrow$) |
|---|---|---|
| Hebbian + Cache (Rae et al., 2018) | 151M | 29.9 |
| Transformer-XL (Dai et al., 2019) | 257M | 18.3 |
| Transformer-XL + Dynamic Eval (Krause et al., 2019) | 257M | 16.4 |
| Compressive Transformer (Rae et al., 2019) | - | 17.1 |
| KNN-LM + Cache (Khandelwal et al., 2019) | 257M | 15.8 |
| Sandwich Transformer (Press et al., 2020a) | 247M | 18.0 |
| Shortformer (Press et al., 2020b) | 247M | 18.2 |
| SegaTransformer-XL (Bai et al., 2021) | 257M | 17.1 |
| Routing Transformer (Roy et al., 2021) | - | 15.8 |
| base LM (Baevski & Auli, 2018) | 247M | 18.7 |
| +GNN | 274M | 16.8 |
| +GNN+$k$NN | 274M | **14.8** |

Table 1: Test perplexity on WikiText-103 dataset.

| Model | # Param | Test ppl ($\downarrow$) |
|---|---|---|
| LSTM+CNN (Jozefowicz et al., 2016) | 1.04B | 30.0 |
| High-Budget MoE (Shazeer et al., 2016) | 5B | 28.0 |
| DynamicConv (Wu et al., 2018) | 0.34B | 26.7 |
| Mesh-Tensorflow (Shazeer et al., 2018) | 4.9B | 24.0 |
| Evolved Transformer (Shazeer et al., 2018) | - | 28.6 |
| Transformer-XL (Dai et al., 2019) | 0.8B | 21.8 |
| Adaptive inputs (base) (Baevski & Auli, 2018) | 0.36B | 25.2 |
| Adaptive inputs (large) (Baevski & Auli, 2018) | 0.46B | 23.9 |
| base LM (Baevski & Auli, 2018) | 1.03B | 23.0 |
| +$k$NN | 1.02B | 22.8 |
| +GNN | 1.05B | 22.7 |
| +GNN+$k$NN | 1.05B | 22.5 |

Table 2: Test perplexity on One Billion Word dataset.

Character (BPC), achieving 1.03 BPC with only 48M parameters, comparable to 18L Transformer-XL with 88M parameters.

| Model | # Param | BPC ($\downarrow$) |
|---|---|---|
| 64L Transformer (Al-Rfou et al., 2019) | 235M | 1.06 |
| 18L Transformer-XL (Dai et al., 2019) | 88M | 1.03 |
| 24L Transformer-XL (Dai et al., 2019) | 277M | 0.99 |
| 24L Transformer-XL + Dynamic Eval (Krause et al., 2019) | 277M | 0.94 |
| Longformer (Beltagy et al., 2020) | 102M | 0.99 |
| Adaptive Transformer (Sukhbaatar et al., 2019) | 209M | 0.98 |
| Compressive Transformer (Rae et al., 2019) | 277M | 0.97 |
| Sandwich Transformer (Press et al., 2020a) | 209M | 0.97 |
| 12L Transformer-XL (Dai et al., 2019) | 41M | 1.06 |
| +$k$NN | 41M | 1.04 |
| +GNN | 48M | 1.04 |
| +GNN+$k$NN | 48M | 1.03 |

Table 3: Bit per Character on the Enwik8 dataset.

# 4 ANALYSIS

## 4.1 COMPLEXITY ANALYSIS

**Space Complexity** In our model, we consider $k$ nearest neighbors for each token $c_i$ in context; the number of nodes in the graph is $k$ times larger than vanilla LM during training. Accordingly, training GNN requires approximately $k$ times larger memory than vanilla LM, since we have to

maintain hidden representations of each node for backward propagation. We propose two strategies to alleviate the space issue: (1) For all datasets, we first train with a smaller $k = 32$, then further finetune the model with a larger $k = 128$; and (2) For datasets with extremely long dependency (e.g., WikiText-103), we truncate the context to a smaller length (e.g., 128) instead of the original longer context (e.g., 3,072) used by vanilla Transformer (Baevski & Auli, 2018). Note that we build GNN model on top of the vanilla Transformer, and the parameters of Transformer are fixed when GNN parameters are being trained. Hence, the GNN could exploit long dependency information learned by Transformer without having to build a large graph with long context. Figure 2(a) shows the comparison of base LM and GNN-LM on GPU memory usage with variant $k$ in WikiText-103.[4]

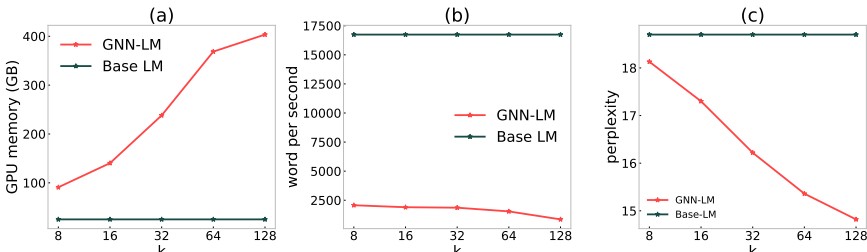

Figure 2: Comparisons between base LM and GNN-LM on WikiText-103 with respect to different $k$. (a) GPU memory usage. (b) Speed (word per second). (c) Test perplexity.

**Time Complexity**   Both GNN and Transformer consist of two basic modules: the feed forward layer and the attention layer. Let $|V|$ be the number of nodes and $|E|$ be the number of edges in the graph. Then the time complexity of the feed forward layer is $\mathcal{O}(|V|)$ and the time complexity of the attention layer is $\mathcal{O}(|E|)$. The GNN model increases $|V|$ by $(l + r + 1)k$ times in the graph, and adds $(l + r + 1)k|V|$ edges to the graph. Note that $|E| = |V|^2$ in Transformer, and thus the increased time complexity is acceptable if $k \ll |V|$ holds. Figure 2(b) shows the comparison between base LM and GNN-LM in speed in WikiText-103. We observe that the speed of GNN-LM is approximately 8 to 20 times slower than the base LM (Baevski & Auli, 2018) with respect to different $k$.

It is worth noting that the overhead of the proposed model comes from $k$NN retrieval, which can be done in advance and thus does not result in time overhead when running the model. Specifically, the time overhead for retrieval comes from two processes: 1) building data indexes with token representations in the train set; 2) collecting nearest neighbors by querying the data indexes. For WikiText-103, building data indexes takes approximately 24 hours on a CPU machine with 64 cores. And querying data indexes for all tokens in train set takes approximately 30 hours.

## 4.2 ABLATION STUDY

**Number of Neighbors per Token**   The number of neighbors per source token (i.e., $k$) significantly influences how much information could be retrieved from the training set. Figure 2(c) shows that test perplexity monotonically decreases when $k$ increases from 8 to 128. This trend implies that even larger improvements can be achieved with a larger value of $k$.

**Neighbor Quality**   We evaluate the quality of $k$NN retrieval by examining whether the target token to predict (i.e., $w_t$) is the same as the token that comes right after the retrieved nearest sequence using the recall metric. Given a sample $\boldsymbol{c}_t = (w_1, w_2, ..., w_{t-1})$ and its $k$NN $\mathcal{N}(\boldsymbol{c}_t) = \{\boldsymbol{c}_{t_1}^{(1)}, ..., \boldsymbol{c}_{t_k}^{(k)}\}$, the quality of $k$NN retrieval is defined by

$$R(\boldsymbol{c}_t) = \sum_{i=1}^{k} \mathbb{1}\left[w_t = w_{t_i}^{(i)}\right], \tag{6}$$

where $w_t$ is the target token to predict at time step $t$, and $w_{t_i}^{(i)}$ is the token that comes right after the $i$-th retrieved neighbor. We calculate and then divide all samples in the WikiText-103 test set by the

---

[4]We note base LM uses a context length of 3,072, while the context length of GNN-LM is 128. We scale up the value of GNN-LM 24 times for fair comparison.

| kNN recall range | [0, 4) | [4, 27) | [27, 137) | [137, 463) | [463, 1024] |
|---|---|---|---|---|---|
| base LM | -7.14 | -3.84 | -2.21 | -1.19 | -0.30 |
| +GNN+$k$NN | -7.15 | -3.46 | -1.71 | -0.80 | -0.21 |
| absolute improvement | -0.01 | 0.38 | 0.50 | 0.39 | 0.09 |
| relative improvement | -0.0% | 10% | 23% | 33% | 32% |

Table 4: Comparison of base LM and GNN-LM in different $k$NN recall buckets. We report average log probabilities within each bucket, and compute the absolute and relative improvement.

recall value into 5 buckets, with each bucket containing around 50k tokens. Results are reported in Table 4. We observe that GNN-$k$NN-LM gains more relative improvements to base LM when the quality of $k$NN retrieval reaches a relatively high level.

**Representation in $k$NN**   We finally study the effect of using different representations in the $k$NN scoring function in Section 2.4. We experiment with two types of representations: (1) from the last layer of Transformer, which is the default setting, and (2) from the last layer of GNN. The model performances with different choices for query and key are reported in Table 5. Results show that using GNN representations for both query and key leads to the best performance. It suggests that GNN learns better representations for context similarity. We also observe that the performance is marginally worse when both query and key are using Transformer representations. Considering that building an additional datastore for GNN representations is computationally intensive, in practice we can directly use Transformer representations (the default setting).

| Query Repres. | Key Repres. | Test ppl ($\downarrow$) |
|---|---|---|
| Transformer | Transformer | 14.82 |
| Transformer | GNN | 15.16 |
| GNN | Transformer | 14.97 |
| GNN | GNN | 14.76 |

Table 5: Test perplexity on WikiText-103 with different representations as query and key.

### 4.3 EXAMPLES

Table 6 presents two examples showing the input and the corresponding extracted three neighbor contexts. The two examples demonstrate that the extracted contexts have a strong connection in semantics to the input, and thus leveraging the neighboring information will benefit model predictions.

| |
|---|
| *Input*: In 2000 Boulter had a guest @-@ **starring** |
| *Extracted 1*: In 2009 , Beghe had a guest @-@ starring role on the television show Californication . |
| *Extracted 2*: had previously worked on Hack , for a guest @-@ starring episode arc on the show . |
| *Extracted 3*: and because of Patrick Stewart 's hilarious guest @-@ starring role as " Number One . " |
| *Input*: Tourism is a vital industry in Manila , **and** |
| *Extracted 1*: a large audience in Mogadishu , and was widely sold prior to the civil war . |
| *Extracted 2*: industry is well established , with Mumbai Port being one of the oldest and most |
| *Extracted 3*: transportation has become a large business in Newark , accounting for more than 17 |

Table 6: Two examples showing the input context and the corresponding extracted three neighbors. The **bold** token is the gold token to predict, and the underlined are the extracted context tokens.

## 5 RELATED WORK

**Language Modeling**   Traditional methods for language modeling use $n$-gram statistics to compute the probability of the next token given the $(n-1)$-gram context (Bahl et al., 1983; Nadas, 1984; Chen & Goodman, 1999). With the development of neural language models (NLMs) (Mikolov et al., 2012),

deep learning based methods begin to dominate the learning paradigm of language modeling. For example, Jozefowicz et al. (2016) built a strong language model by combining the LSTM (Schuster & Paliwal, 1997) model and the CNN structure; Melis et al. (2017); Merity et al. (2017) applied a variety of regularizations to LSTMs and achieved state-of-the-art results; Baevski & Auli (2018) proposed adaptive input embeddings, which can improve performance while drastically reducing the number of model parameters. On top of Transformer (Vaswani et al., 2017), considerable efforts have been devoted to building stronger and more efficient language models (Shazeer et al., 2018; Dai et al., 2019; Beltagy et al., 2020; Press et al., 2020b;a). BERT (Devlin et al., 2018) proposed the Masked Language Modeling (MLM) pretraining paradigm to train a deep bidirectional Transformer model; RoBERTa (Liu et al., 2019) removed the Next Sentence Prediction (NSP) task in BERT; XLNet (Yang et al., 2019) generalized BERT pretraining to the autoregressive manner; Span-level BERTs (Lewis et al., 2019; Song et al., 2019; Joshi et al., 2020) introduced span-level masks rather than just relying on token-level masks. ELECTRA (Clark et al., 2020) proposed to detect token replacement as opposed to token generation, improving both the efficiency and effectiveness of pretraining. Sun et al. (2021) extends BERT to accommodate glyph information.

**Graph Neural Networks**    Graph neural networks (GNNs) capture the dependencies and relations between nodes connected with edges, which propagate features across nodes layer by layer (Scarselli et al., 2008; Kipf & Welling, 2016; Hamilton et al., 2017). GNNs have demonstrated effectiveness in a wide variety of tasks in natural language processing such as text classification (Yao et al., 2019; Lin et al., 2021), machine translation (Bastings et al., 2017), question answering (Song et al., 2018; De Cao et al., 2018), recommendation (Wu et al., 2019) and information extraction (Li et al., 2020a). For example, Guo et al. (2019) proposed Star Transformer, a Transformer backbone but replaces the fully-connected structure in self-attention with a star-like topology. Ye et al. (2019) adopted a fine-to-coarse attention mechanism on multi-scale spans via binary partitioning (BP). Li et al. (2020b) proposed to learn word connections specific to the input via reinforcement learning.

**Retrieval-augmented Models**    Retrieving contexts from another corpus as additional information improves the model's robustness towards infrequent data points. A typical application of retrieval-augmented models is open-domain question answering, which solicits related passages from a large open-domain database to answer a given question. The dominant approach is to cache dense representations of the passages and retrieve the closest ones to the input during inference (Lewis et al., 2020b; Karpukhin et al., 2020; Xiong et al., 2020; Lee et al., 2020; Li et al., 2020b). Lewis et al. (2020a) proposed to first extract a set of related texts and condition on them to generate the target text. allowing for strong zero-shot performance. Besides open-domain QA, other tasks such as language modeling (Khandelwal et al., 2019; Guu et al., 2020), machine translation (Zhang et al., 2018; Tu et al., 2018; Jitao et al., 2020), text classification (Lin et al., 2021), and task-oriented dialog generation (Fan et al., 2020; Thulke et al., 2021) also benefit from the additionally retrieved information. For example, Khandelwal et al. (2019) retrieved $k$ nearest neighbors from a large-scale unannotated corpus and interpolates with the decoded sentence for language modeling. Khandelwal et al. (2020); Meng et al. (2021) retrieved $k$NNs from the parallel translation corpus to augment the machine translation outputs. However, these methods retrieve related texts *independently*.

## 6    CONCLUSION AND FUTURE WORK

In this work, we propose GNN-LM, a new paradigm for language modeling that extends vanilla neural language model by allowing to reference similar contexts in the entire training corpus. High dimensional token representations are used to retrieve $k$ nearest neighbors of the input context as reference. We build a directed heterogeneous graph for each input context, where nodes are tokens from either the input context or the retrieved neighbor contexts, and edges represent connections between tokens. Graph neural networks are then leveraged to aggregate information from the retrieved contexts to decode the next token. Experimental results show that our proposed method outperforms strong baselines in standard benchmark datasets, and by combining with $k$NN LM, we are able to achieve state-of-the-art results on WikiText-103. In future work, we will consider improving efficiency for building the graph and retrieving nearest neighbors.

## ACKNOWLEDGEMENT

This work is supported by the Science and Technology Innovation 2030 - "New Generation Artificial Intelligence" Major Project (No. 2021ZD0110201) and the Key R & D Projects of the Ministry of Science and Technology (2020YFC0832500). We would like to thank anonymous reviewers for their comments and suggestions.

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
