# OpenReview forum: "GNN-LM: Language Modeling based on Global Contexts via GNN"
_ICLR.cc/2022/Conference — ICLR 2022 Spotlight_

### Official Review · Reviewer_BmDz · 2021-11-01

**Correctness:** 3
**Technical Novelty And Significance:** 2
**Empirical Novelty And Significance:** 3
**Recommendation:** 6
**Confidence:** 3

**Details Of Ethics Concerns:**

I do not have any concerns regarding to the ethics.

**Main Review:**

The work is among the studies of enhancing language generation by context retrieval, and the new idea is modeling the retrieved neighbor contexts through a graph neural network.  My concerns lie in the following aspects:

1. Comparison: it seems that the proposed model is compared with a retrieval-augmented baseline  only on WikiText-103. Why the comparison is not conducted on the other two benchmarks? Is it because there are no available results on the two other dataset? In this case, can you implement the model and make the comparison? Moreover, there have been many retrieval-augmented models, though some of them are not applied to LM tasks in the initial papers, then can you make adaptations and compare with them as well? Since the major contribution of the work is the GNN module, it would be important to demonstrate how useful the component is.

2. Clarity: I feel confused at several places during the review process: (1) Eq.(5), the text below explains that f(\cdot) is a neural language model, but afterwards in Table 5, it seems that GNN is a better choice. What is the exact setting for this equation? (2) The first paragraph of Section 3, the authors say "we retrieve k=1024 nearest neighbors for each source token, among them the top 128 neighbors are used in graph, and all of them are used in computing the kNN-baed probability", then can I understand as the graph only involves 128 tokens? If it is, given GNN is better than Transformer in Table 5, how to calculate f(\cdot) in Eq.(5) for other tokens? (3) how do you train the transformer and the GNN？There are some comments above Figure 2, but do you train the two components iteratively? If yes, how do you set up the training procedure? (e.g., how many iterations do you do? how to warm up the parameters of the GNN?) (4) Eq. (6) is really confusing. You only compare w_t with the i-th token of w_{t_i}? Why don't you use soft similarity (e.g., Cosine)?

**Summary Of The Paper:**

The paper presents a GNN based language model where neighbor contexts are retrieved, encoded via a graph neural network, and used to enhance generation. Evaluation on three benchmarks indicates that the proposed approach can outperform a bunch of baseline models.

Contributions:
1. a new retrieval-augmented language model implemented via GNN techniques.
2. improvements over state-of-the-art models on three benchmarks.

**Summary Of The Review:**

1. Relatively incremental technical contribution to the community.
2. Relatively weak comparison with baselines.
3. Many vague points that impede us from understanding (reproducing) the work.

——————————————————————————————————————————————————————————————

The authors' response answered most of my questions. though I still feel that the technical contribution is big enough. I slightly raise my score accordingly.

---

> ### Author Response · Authors · 2021-11-19
> **Response to Reviewer BmDz**
>
> Thanks for the comments. We would like to clarify the raised concerns.
>
> &nbsp;
>
> **Q1.**  “Why the comparison with KNN-LM is not conducted on the other two benchmarks? …”
>
> **A1.** We re-implemented KNN-LM on Enwiki8 and OneBillion datasets since the author of KNN-LM did not report the results. The full experimental results are shown below. We observe that the proposed model achieves the best performance, better than KNN model on Enwiki8 and OneBillion datasets. We have added the comparison with KNN-LM on the other two benchmarks in the updated paper.
>
> | EnWik8 Dataset	|		BPC |
> | :- | |
> |**Base LM**  	|		1.06|
> |**Base LM + KNN** 	|	1.04|
> |**Base LM + GNN** 	|	1.04|
> |**Base LM + GNN+KNN**| 1.03|
>
> |One Billion Dataset	|		PPL |
> | :- | |
> |**Base LM**  	|		23.0|
> |**Base LM + KNN** 	|	22.8|
> |**Base LM + GNN** 	|	22.7|
> |**Base LM + GNN+KNN** |22.5|
>
> &nbsp;
>
> **Q2.** there have been many retrieval-augmented models, though some of them are not applied to LM tasks in the initial papers, then can you make adaptations and compare with them as well?
>
> **A2.**  Thanks for the comment. In this paper, we need to obtain token representations given the input context.  Other retrieval-augmented models [1, 2, 3] often focus on sentence/document representations in large corpora (encode a document/sentence to a fixed-length vector).  Token representations in these retrieval-augmented models are not well studied and thus cannot be readily extended to this work.
>
> [1] Boosting Neural Machine Translation with Similar Translations. Xu et al. (ACL 2020)
>
> [2] REALM: Retrieval-Augmented Language Model Pre-Training. Guu et al. (ICML 2020)
>
> [3] Retrieval-Augmented Generation for Knowledge-Intensive NLP Tasks. Piktus et al. (NeurIPS 2020)
>
> &nbsp;
>
> **Q3.** Eq.(5), the text below explains that f(\cdot) is a neural language model, but afterwards in Table 5, it seems that GNN is a better choice. What is the exact setting for this equation?
>
> **A3.** Sorry for the confusion. In Table 5, we explore the effect of using different representations (e.g., Transformer-based LM, GNN-LM) for constructing the datastore for ablation study purpose, and we found that GNN-LM is a better choice. But, in practice, Transformer-based LM is significantly faster than GNN-LM for KNN selection,  we chose to use Transformer-based LM, the results of which were reported in Table 1-4.
>
> &nbsp;
>
> **Q4.** Details of “retrieve k=1024 nearest neighbors for each source token, among them the top 128 neighbors are used in graph”
>
> **A4.** Sorry for the confusion. For Base LM + GNN+KNN, we need to compute both the KNN and GNN probabilities. For KNN, we used 1,024 nearest neighbors. For GNN, we ony take 128 neighbors to build nearest graphs datastore for computation efficiency. These 128 neighbors are the top ones selected from the 1,024 neighbors from KNN. We have made this point clearer in the updated version.
>
> &nbsp;
>
> **Q5.** how do you train the transformer and the GNN？
>
> **A5.** Sorry for the confusion. Please refer to Figure 1 or the beginning paragraph of Page 7 for more details. Before training GNN, we first pretrain a Transformer-LM for finding nearest neighbors. Then we fix parameters in the Transformer-LM for training GNN. We train the Transformer-LM and GNN for one time.
>
> &nbsp;
>
> **Q6.** Eq. (6) is really confusing.
>
> **A6.** Sorry for the confusion. The basic idea is that, we wish to see whether the target token to predict (i.e., $w_t$) is the same as the token that comes right after the retrieved nearest sequence. If they are the same, it means that the retrieved nearest sequence can provide important and correct evidence for the upcoming word prediction.
> Specifically, $w_{t_i}^{(i)}$ denotes the the ${t_i}$-th (i.e., last) token of the $i$-th nearest neighbor, where $t_i$ denotes the length of $i$-th neighbor. The recall metric in Eq. 6 is just comparing whether $w_t$ (the target token to predict) is the same as the $w^{(i)}_{t_i}$ (the token that comes right after the $i$-th retrieved nearest sequence).
>
> &nbsp;
>
>
> **Q7.** You only compare w_t with the i-th token of w_{t_i}? Why don't you use soft similarity (e.g., Cosine)?
>
> **A7.** Firstly, hard similarity is more precise to evaluate the quality of retrieved sequences, simply comparing whether the target token to predict (i.e., $w_t$) is the same as the token that comes right after the retrieved nearest sequence. Secondly, it is more robust as it does not rely on any form of vector representations to compute the soft similarity. It is worth noting that the hard strategy has been widely adopted for KNN quality evaluation (directly implemented in the FAISS package.).

---

### Official Review · Reviewer_eytL · 2021-11-03

**Correctness:** 4
**Technical Novelty And Significance:** 3
**Empirical Novelty And Significance:** 3
**Recommendation:** 10
**Confidence:** 4

**Main Review:**

I would recommend  accepting the paper based on the novel idea to build heterogeneous graph to do LM and the impressive model performance on PPL.  This work further extends KNN-LM to utilize not next tokens but all neighborhood information to get global context. This good combination of GNN and LM can be valuable to the community. The improvement on PPL also shows the importance to use global knowledge.

The heterogeneous GNN is standard. I would like to know if the authors have thought about designing specific graph structure or avoid inter-context edge when building the graph or considering "is" and "are" are the same node when building the graph in Fig 1.

I would consider KNN-LM is a special version of GNN-LM. Then why adding KNN can greatly further improve the model performance?

**Summary Of The Paper:**

This work build a novel GNN-LM to do language modeling by using global context information. The proposed model is novel and quite different from previous LM structures. This work in my view draws the connection between traditional n-gram language model and neural language model. The overall performance is quite impressive in all standard LM datasets. Extensive ablation study is conducted to understand the model.

**Summary Of The Review:**

Overall, base on the novel idea of creating global context graph, GNN-LM and show the significant improvements on all LM datasets, I would like to recommend accept this paper.

---

> ### Author Response · Authors · 2021-11-19
> **Response to Reviewer eytL**
>
> Thank you for the encouraging comments. We would like to explain the questions raised by the reviewer below.
>
> &nbsp;
>
> **Q1.** The heterogeneous GNN is standard. I would like to know if the authors have thought about designing specific graph structure or avoid inter-context edge when building the graph or considering "is" and "are" are the same node when building the graph in Fig 1.
>
> **A1.** The reasons why we set edges/nodes to two different types in graph construction stage are as follows: 1) for edges, we cannot add the same type edges in the reverse direction because of data leakage. Two types of edges are used to differentiate the information from intra-context tokens and inter-context tokens. 2) for nodes, the type $a_o$ and $a_n$ are naturally inequivalent. We will explore more on designing specific graph structures in the future work.
>
> &nbsp;
>
> **Q2.** I would consider KNN-LM is a special version of GNN-LM. Then why adding KNN can greatly further improve the model performance?
>
> **A2.** Sorry for the confusion. GNN-LM is an extension to softmax-based neural LM and thus faces the issue that softmax does not have the capacity to express the true data distribution (the softmax bottleneck issue mentioned in [1]). In contrast, KNN does not limited by the softmax bottleneck and aggregates probability mass for each vocabulary item across all its occurrences in the retrieved targets. Therefore, KNN and GNN-LM actually can complement each other, leading to better final performance.
>
> &nbsp;
>
> **Reference:**
>
> [1] Breaking the softmax bottleneck: A high-rank RNN language model. Yang et al. (ICLR 2018)

---

> > ### Comment · Reviewer_eytL · 2021-11-29
> > **Response to rebuttal**
> >
> > I have read the author's rebuttal. The author address all my question.

---

### Official Review · Reviewer_54po · 2021-11-04

**Correctness:** 4
**Technical Novelty And Significance:** 3
**Empirical Novelty And Significance:** 4
**Recommendation:** 8
**Confidence:** 3

**Main Review:**

Strength:
- The authors propose an interesting hypothesis that referring to the training data could be helpful for language modeling, and they showed that the method is able to make considerable improvements over the vanilla LM.
- The method achieves the new SOTA on Wiki103, which is impressive

Weaknesses:
- The paper lacks a discussion part about the actual overhead for retrieval and the time overhead for running the model seems to be significant (8-20X slower).
- Some hyperparameters seem to be chosen quite arbitrarily like l and r, maybe the authors could provide more insights as in why they chose such a small number. Does it affect the efficiency or the performance much?

**Summary Of The Paper:**

This paper proposes a method to model original texts and similar texts in a graph structure for language modeling with graph neural networks. In the graph, the nodes are texts or similar contexts and the edges are connections between the nodes. The new model achieves the new state-of-the-art on WikiText-103 and shows substantial improvements over other language modeling datasets such as One Billion Word and Enwiki8 datasets .

**Summary Of The Review:**

The paper proposes a retrieval augmentation to the language models and use GNNs on top of a graph structure of the original input and its neighbors and shows that it consistently improves over the vanilla lm model. The empirical results are good and the authors provide examples showing that the retrieved examples indeed help prediction in language models. The only concern is that through modeling additional neighboring contexts, the method introduces significant overhead in running time.

---

> ### Author Response · Authors · 2021-11-19
> **Response to Reviewer 54po**
>
> We would like to thank the reviewer for the encouraging comments and address the concerns below.
>
> &nbsp;
>
> **Q1.** The paper lacks a discussion part about the actual overhead for retrieval and the time overhead for running the model seems to be significant (8-20X slower).
>
> **A1.** Sorry for missing these details. Regarding retrieval, it can be done in advance and thus does not result in time overhead when running the model.  Specifically,  the time overhead for retrieval comes from two processes: 1) building data indexes with token representations in the train set; 2) collecting nearest neighbors by querying the data indexes.  For WikiText-103, building data indexes takes approximately 24 hours on a CPU machine with 64 cores. And querying data indexes for all tokens in train set takes approximately 30 hours.
>
> For the running time of our model, the GNN structures lead to an about 8 times slower than the vanilla model because we need to refer to way more context data using GNN. We will definitely work on a more efficient version in the future work. We have added these details in the updated paper.
>
> &nbsp;
>
> **Q2.** Some hyperparameters seem to be chosen quite arbitrarily like l and r, maybe the authors could provide more insights as in why they chose such a small number. Does it affect the efficiency or the performance much?
>
> **A2.** Sorry for missing details regarding hyper-parameter selection. The reasons why we use small values of $l$ and $r$ are as follows: 1) during experiments, we find that using shallow (i.e., 3) GNN layers and adding $r_\text{intra}$ edges between adjacent tokens can alleviate overfitting. Since a 3-layer GNN only aggregates information from 3-hop nodes in the graph, using larger $l$ and $r$ have no influence on GNN representations. 2) the larger values of $l$ and $r$ would not affect the performance but result in greater time and space complexity. It makes the model infeasible to train. We will clarify this point in the updated version.

---

### Decision · Program_Chairs · 2022-01-20

**Decision:**

Accept (Spotlight)

**Comment:**

This paper introduces a new type of language model, the GNN-LM, which uses a graph neural network to allow a language model to reference similar contexts in the training corpus in addition to the input context.  The empirical results are good, and the model sets a new SOTA on the benchmark Wikitext-103 corpus, as well as improving over strong baselines on two other language modeling datasets (enwiki8 and Billion Word Benchmark).  The main drawback, as noted by one reviewer, is the computational expense of the method with significant slowdowns compared to the baseline.

Two reviewers voted strong accept, with a third raising several concerns.  The largest concern was the lack of comparison to prior work, especially prior retrieval based methods on two datasets.  The authors responded with an ablation study comparing their method to KNN-LM and showed their proposed GNN-LM performs better.  Other concerns raised by the reviewer were the paper's lack of clarity (the authors should address the reviewers questions during the next revision) and incremental technical contribution.  Another reviewer highlighted the paper's novelty, and this AC agrees it is sufficient for publication.

Overall, the method is an interesting, if expensive, extension of retrieval based language models, and the empirical results support its effectiveness.